# EXPLORING ROUTING STRATEGIES FOR MULTILINGUAL MIXTURE-OF-EXPERTS MODELS

## ABSTRACT

Sparsely-Gated Mixture-of-Experts (MoE) has been a successful approach for scaling multilingual translation models to billions of parameters without a proportional increase in training computation. These models, however, are prohibitively large for serving deployment and there is no easy way to extract a sub-network to decode for a particular language pair. This work proposes improved strategies to route MoE models by tasks instead of tokens, thus enabling separation of network structures at decoding time while enjoying the benefits of scale and task sharing at training time. We compare routing strategies at multiple levels (token, sentence, task) in both, the encoder and the decoder, and conduct extensive experiments on two benchmarks: the public WMT dataset of 30 language pairs and an in-house web-scale dataset of 200 language pairs. On WMT, with a Transformer base model with 32 experts, our task-level MoE outperforms the best performing token-level MoE model by +1.0 BLEU on average over all language pairs. When scaling up to Transformer big model with 128 experts on the large-scale massively multilingual benchmark, our task-level MoE is competitive with token-level MoE while being able to reduce the decoder model size by a factor of 32.34 and increase peak throughput by 2.6 times at inference.

## 1 INTRODUCTION

Scaling up neural network models has recently received great attention, given the significant quality improvements in a variety of areas such as natural language understanding (Raffel et al., 2019; Brown et al., 2020) and multilingual machine translation (Huang et al., 2019; Lepikhin et al., 2020).

While training massive models on large amounts of data can almost guarantee improved quality, there are two factors affecting their practicality and applicability: (1) *training efficiency* and (2) *inference efficiency*. Large dense models are often prohibitively compute-intensive to train, with some models requiring TFlops-days of compute (Brown et al., 2020). A recent line of work has proposed sparsely-gated Mixture-of-Experts (MoE) layers as an efficient alternative to dense models (Shazeer et al., 2017; Lepikhin et al., 2020; Riabinin & Gusev, 2020) in order to address *training efficiency* limitations. In a vanilla sparsely-gated MoE model each token of the input sequence activates a different subset of the experts, hence the computation cost per token becomes only proportional to the size of the activated sub-network. However, they fail to meet requirements on *inference efficiency*.

Consider a long sequence where each token of the sequence activates a disjoint subset of available experts. From a practical standpoint, the inference trace of the full sequence spans several experts independently for every token, resulting in an independent pathway for each token. Although this is a desired property adding flexibility to the model and increasing its capacity, it becomes prohibitive for inference for the following reasons: The model parameters in these large models are beyond the memory limit of a single accelerator, and require model parallelism to shard them across a cluster of devices during inference. For models with MoE Layers, the input token would be dynamically routed to different experts allocated to different devices. This further adds communication cost across devices to the overall serving cost. Moreover, due to the sequential nature of the auto-regressive decoding (Kasai et al., 2020; Chen et al., 2018), the added communication cost from model parallel decoders gets multiplied by the number of decoding steps. To add to this, serving MoE models efficiently requires batching a large number of input tokens together, otherwise only a subset of the MoE network will be activated leading to device under-utilization.

In this work, we study the *inference efficiency* of sparsely gated MoE models while taking into account the characteristics of the intended application, Multilingual Neural Machine Translation (MNMT). MNMT is an inherently multi-task learning problem, aimed at building a single neural network for translating multiple language pairs simultaneously. In a MNMT model, the extent to which parameters are shared across languages determines the magnitude of positive transfer (Baldwin & Ford, 1988) and conversely task interference due to the capacity bottleneck (Arivazhagan et al., 2019). In an ideal scenario, we would want to efficiently train a single large MNMT model maximizing transfer while expanding the capacity bottleneck; at the same time, we would like to enjoy the benefits of sparsely activated sub-networks per-task at inference time, i.e. extracting out a sub-network from the model to decode for a particular language pair to actualize *inference efficiency*.

We propose routing algorithms for MoE models with affordable serving costs. While vanilla MoEs route each sub-word token in the input to its preferred experts, we explore alternative routing strategies that leverage global task level information to route all tokens corresponding to a particular task collectively to the same set of experts. While this strategy could be perceived to be restrictive for parameter sharing across tasks, we empirically demonstrate that routing based on task boundaries performs better when applied to MNMT. During training, we mix the inputs from different tasks in the same batch in order to learn the routing network and encourage positive transfer among the tasks. During inference, we decode different tasks separately and only load the subset of experts associated with the corresponding task.

We compare our method with multilingual baselines and find that we achieve significant gains on two benchmarks: a multilingual WMT task with comparable inference cost (+3.59 BLEU), described in Section 4, and a large internal dataset (+3.6 BLEU), in Section 4.3.2). We see that the gains are comparable with conventional position-wise Mixture-of-Expert models while utilizing decoders with only a fraction (6.25% and 1.56%) of their serving cost. We discuss the trade-offs of these different methods in Section 3.2. In Section 4.3.4, we analyze the routing decisions made in MoE models and motivate our method.

## 2    SCALING TRANSFORMERS WITH MIXTURE-OF-EXPERTS

The Transformer (Vaswani et al., 2017) architecture is a popular model used for neural machine translation and other natural language understanding problems. In sequence-to-sequence problems (of which neural machine translation is one example), the model consists of a separate encoder and decoder, each of which contains multiple Transformer layers. For further details on Transformers, we refer the reader to the original paper (Vaswani et al., 2017).

We use the Mixture-of-Experts Transformer models used by Lepikhin et al. (2020), where the MoE layers for the Transformers consist of $E$ feed-forward networks (FFN), such that ($\text{FFN}_1 \dots \text{FFN}_E$).

$$\text{FFN}_e(x_s) = wo_e \cdot \text{ReLU}(wi_e \cdot x_s)$$

$$y_s = \sum_{e=1}^{E} \mathcal{G}_{s,e} \cdot \text{FFN}_e(x_s)$$

Here, $x_s$ is the input token at position $s$ to the MoE layer and each $\text{FFN}_e$ is a two layer neural network using a ReLU activation function. $wi_e$ and $wo_e$ are the input and output projection weights of the $e$-th expert. Finally, $\mathcal{G}_{s,E}$ is vector computed by the gating network. For each expert, most values of this vector are zeros, one value being positive. We use this vector to route the token to a select few experts. The entries chosen from $\mathcal{G}_{s,E}$ determine how much the expert contributes to the final output $y_s$. Note that, in this work we choose the top 2 weight experts for each example to be comparable with the prior work.

The gating network $\mathcal{G}_{s,E}$ must be considered carefully for efficiency purposes: (1) the utilization of experts must be balanced and (2) the function must be efficient to implement at scale. For a more thorough discussion of MoE transformer, we direct the reader to Lepikhin et al. (2020).

## 3    METHODS

In this section we describe our candidate routing strategies in the context of MNMT and discuss their trade-offs from the perspective of the training and inference efficiency. It is known that multi-

lingual models learn different overlapping representations depending on the language - this is true for both dense (Wu & Dredze, 2019; Tiedemann, 2018; Tan et al., 2019; Östling & Tiedemann, 2016; Kudugunta et al., 2019) and sparse models (Section 4.3.4). Therefore we propose changing the routing algorithm GATE($x_s$) of MoEs to choose different experts using more natural separations.

### 3.1 ROUTING STRATEGIES

Given the sequential nature of the multilingual machine translation task, the routing decisions can be made at three different granularities, from bottom up (i) token-level, (ii) sentence-level and (iii) task-level, as detailed below.

- **Token-level Routing:** This is the baseline discussed in Section 2 where each token is routed independently.
- **Sentence-level Routing:** Each sequence (sentence), and all tokens that form the sequence, are routed to the same expert. We change the routing algorithm to select experts by sentence representation, calculated by taking the average token representations in a given sentence.
- **Task-level Routing:** We select experts by task boundaries as opposed to making input-level decisions. In the context of MNMT, these task boundaries can either be defined by the target language (French-to-English and German-to-English are the same task) or the language pair (French-to-English and German-to-English are different tasks).

$$\mathcal{G}_{s,E} = \text{GATE}(\frac{1}{S}\sum_{s=1}^{S} x_s) \quad \text{(Sentence-level routing)} \tag{1}$$

$$\mathcal{G}_{s,E} = \text{GATE}(\text{task\_id}_s) \quad \text{(Task-level routing)} \tag{2}$$

We further illustrate the difference in Figure 1, in token-based MoE models (Figure 1a), tokens from each example are routed to different experts, whereas in task-level MoE models (Figure 1b), tokens may be routed to the same expert based on task.

### 3.2 INFERENCE IMPLICATIONS OF ROUTING STRATEGIES

While the MoE models discussed in (Shazeer et al., 2017; Lepikhin et al., 2020) train quickly relative to the number of parameters in terms of the wall-clock time, they are expensive to serve.

Consider a MoE with 512 experts and 50B parameters (Lepikhin et al., 2020). When employing token-level routing, each token can be independently routed to a different set of experts during inference. Given that the entire model is too large to load into memory on a single accelerator, the two potential solutions to utilize this model for inference are: (i) Loading experts dynamically from host to device depending on routing decisions, or (ii) Utilizing model-parallelism over multiple accelerators for serving. While the first solution incurs heavy host-device communication costs, the second introduces significantly inter-device communication overhead.

Another practical approach to serve a large MoE would require model compression via quantization, pruning or distillation (Cheng et al., 2017). While the first two strategies haven't been explored in the context of conditional computation, distillation (Hinton et al., 2015; Kim & Rush, 2016) has been found to introduce undesirable artifacts into the student model (Freitag et al., 2019; Bogoychev & Sennrich, 2019) in the context of NMT. On the other hand, if we limit the number of experts available to every task in the model to a small fraction of the total available capacity, it is possible to extract task-specific models for serving, alleviating the need for complex serving strategies or compression. Since decoding time complexity for auto-regressive seq2seq models is dominated by the decoder (Kasai et al., 2020), we can also pursue a hybrid strategy where the encoder utilizes more expensive routing strategies while the decoder of the model utilizes simpler and efficient routing.

We do note, however, that MoE models that route purely by task boundaries are slower to train due to load balancing considerations. All examples in the input batch belonging to the same task would route to the same set of experts, possibly leading to some experts bearing a significant amount of the load. Balancing between these inference and training time trade-offs merits further exploration.

Summarizing the *effective* decoding cost of the MoE models utilizing different routing strategies:

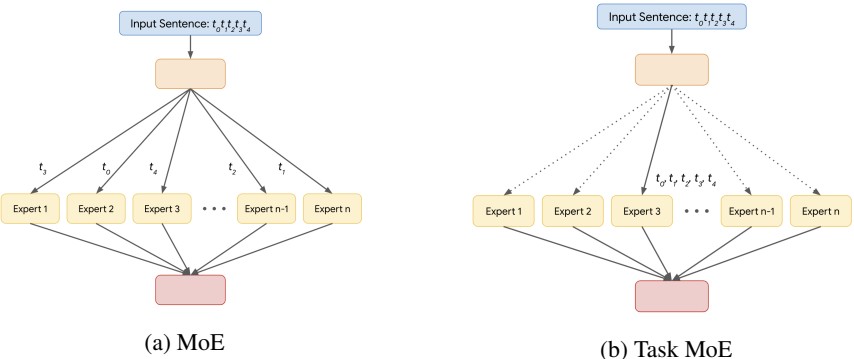

(a) MoE  (b) Task MoE

Figure 1: Different tokens are routed to different experts in token-based MoE models(a), whereas they may be routed to the same expert based on task or some other prior in task-base MoE (b).

- **Token/Sentence level routing**: The routing decisions are made dynamically. Assuming each token/sentence makes disjoint choices, the server needs to load all $E$ experts.

- **Task-level routing**: Tokens corresponding to each input sentence are routed to the same experts statically. The server only needs to pre-load $K$ experts (assuming top-$K$ routing).

## 4 EXPERIMENTS

We compare routing strategies at multiple levels in both, the encoder and the decoder, by conducting extensive experiments on two benchmarks: the public WMT dataset with 30 language pairs (Section 4.1) and an in-house web-scale dataset with 200 language pairs (Section 4.3).

### 4.1 SETUP FOR WMT EXPERIMENTS

For our experiments, we use parallel training and evaluation data from the WMT corpus and adopt the setup used by Siddhant et al. (2020) with 15 languages, to and from English. Full training data details may be found in Table 2 in the Appendix. The amount of data ranges from more than 60 million sentence pairs in en-cs translation direction (en-cs) to roughly 150k sentence pairs for en-gu.

We use a temperature based data sampling strategy to train our models, similar to the strategy used to train the multilingual models in Arivazhagan et al. (2019): if $p_L$ is the probability that a sentence in the corpus belongs to language pair $L$, we sample from a distribution where the probability of sampling from $L$ is proportional to $p_L^{\frac{1}{T}}$. All the experiments in this paper are performed on a model trained with a sampling temperature $T = 5$.

We use the 142M Transformer Base (Vaswani et al., 2017) architecture (or enhanced versions of it with MoE layers) for all of our experiments with WMT. Our models are optimized using Adafactor (Shazeer & Stern, 2018) with momentum factorization and a per-parameter norm clipping threshold of 1.0. We followed a learning rate of 3.0, with 40K warm-up steps for the schedule, which is decayed with the inverse square root of the number of training steps after warm-up. BLEU scores presented in this paper are calculated using SacreBLEU Post (2018) on the WMT test sets.

**Multilingual baseline:** We train a Transformer Base model and a Transformer Big on this dataset as our multilingual dense baselines. We share all parameters across language pairs, including the softmax layer and input/output word embeddings. We use a 64k token Sentence Piece vocabulary (Kudo & Richardson, 2018). The vocabulary is shared on both the encoder and decoder side. Each sentence pair has a `<2xx>` token pre-pended to the source sentence to indicate the target language, following Johnson et al. (2017).

**Mixture of Experts Models:** For MoE models, we replace the feed forward network (FFN) of alternate layers of the Transformer with a set of identical FFN experts as depicted in Figure 1a.

| System | Routing Granularity | | Effective No. of Parameters | | | BLEU | | | | |
|---|---|---|---|---|---|---|---|---|---|---|
| | Encoder | Decoder | Train All | Serving All | Serving Dec. | Average | xx2en | en2xx | High | Low |
| Bilingual Baselines | - | - | 142M | 142M | 25M | 21.01 | 21.81 | 18.9 | 28.15 | 11.81 |
| Multilingual Transformer-Base | - | - | 142M | 142M | 25M | 20.03 | 23.69 | 17.5 | 23.25 | 15.88 |
| Multilingual Transformer-Big | - | - | 473M | 473M | 151M | **23.84** | 26.09 | **22.03** | 27.69 | **18.89** |
| Token-level MoE – 32 experts | Token | Token | 533M | 533M | 221M | 22.58 | 24.91 | 20.35 | 27.49 | 16.28 |
| Sentence-level MoE – 32 expert | Sentence | Sentence | 533M | 533M | 221M | 19.87 | 24.05 | 16.83 | 22.56 | 16.14 |
| Task-level MoE – 32 experts | Language Pair | Language Pair | 533M | 155M | 32M | 21.40 | 25.21 | 16.94 | 23.37 | 17.34 |
| | Target | Target | | 155M | 32M | 22.86 | 25.62 | 20.19 | 27.21 | 17.3 |
| | Language Pair | Token | | 338M | 221M | 22.44 | 25.58 | 20.34 | 26.85 | 16.79 |
| | Target | Token | | 338M | 221M | 22.33 | 24.47 | 20.44 | 26.82 | 16.55 |
| | Token | Language Pair | | 338M | 32M | 23.03 | **26.16** | 20.28 | 27.23 | **17.62** |
| | Token | Target | | 338M | 32M | **23.62** | 25.95 | **21.09** | **28.48** | 17.37 |

Table 1: **Routing strategies for Mixture-of-Experts (MoE) models** – We compare routing experts by either tokens, sentence representations, or tasks (using either language pairs or target languages). For task-level MoE, routing can also be different between encoder and decoder. For results, *Average* is the average results of all language pairs, whereas *xx2en* and *en2xx* are the averages of translations into and from English respectively. *High* indicates high-resource language pairs ($> 1$ million sentence pairs) while *Low* is for low-resource language pairs ($< 1$ million sentence pairs).

For brevity, we provide aggregate BLEU scores in Section 4.2 . We provide the full individual BLEU scores in the Appendix A.3, along with bilingual baselines. In addition, we provide the number of parameters for different components of our models in Appendix A.4.

## 4.2 COMPARISON OF DIFFERENT ROUTING STRATEGIES ON WMT

We compare the token-level, sentence-level and task-level routing strategies discussed in Section 3 at identical network size (32 experts, 533M parameters). The results are presented in Table 1. In general, we find that all types of task-level routing performs better than token-level routing. We see that using sentence representations to route examples (Sentence-level MoE - 32 experts) performs much worse, so we do not conduct further experiments on this setting.

When we use Task MoE on both the encoder and the decoder (Task-level MoE - 32 experts: Target/Target), we see consistent gains across the board. To investigate this further, we trained a model that has (a) Token MoE on the encoder and Task MoE on the decoder (Task-level MoE - 32 experts: Token/Target or Token/Language Pair) and (b) Task MoE on the encoder and Token MoE on the decoder (Task-level MoE - 32 experts: Target/Token or Language Pair/Token). In Table 1 we see that using strategy (a) works the best, whether we choose to route by the target language or the language pair. In Section 4.3.4, we discuss these observations further.

Overall we find that using Task MoE only on the decoder (Task-level MoE 32 experts: Token/Target) works the best, with gains of 1 BLEU over Token MoE. These gains are consistent across xx2en language pairs, en2xx language pairs, high resource languages (more than 1 million sentence pairs), low resource languages and the 2 zero shot pairs.

While the MoE models considered outperform bilingual and multilingual Transformer-Base baselines with comparable inference cost, they are slight outperformed by the multilingual Transformer-Big by 0.2 BLEU on average. Note that Transformer-Big incurs much higher decoding cost. We measured our task-level MoE achieved 8.4x (338k vs 40.3k tokens/sec) higher peak decoding throughput. However, we reiterate that the motivation behind scaling sparsely is to increase capacity with little overhead while remaining competitive with dense models - for example, while it is feasible to train a 473M parameter model (with 8x inference cost), training a much larger dense models to say, 13B model (the size of our scaled up MoE model), is prohibitively slow and expensive.

## 4.3 SCALING UP TO MASSIVELY MULTILINGUAL, MASSIVE MT (M4)

We now scale our results up to a larger internal dataset with over 200 language pairs, while also scaling the number of parameters to beyond 10 billion weights. In addition, we look more closely at the gating decisions made by these sparse models and discuss their implications.

### 4.3.1 EXPERIMENTAL SETUP

**Data:** We use an in-house training corpus generated by crawling and extracting parallel sentences from the web (Uszkoreit et al., 2010). This dataset has 204 direct language pairs (102 languages

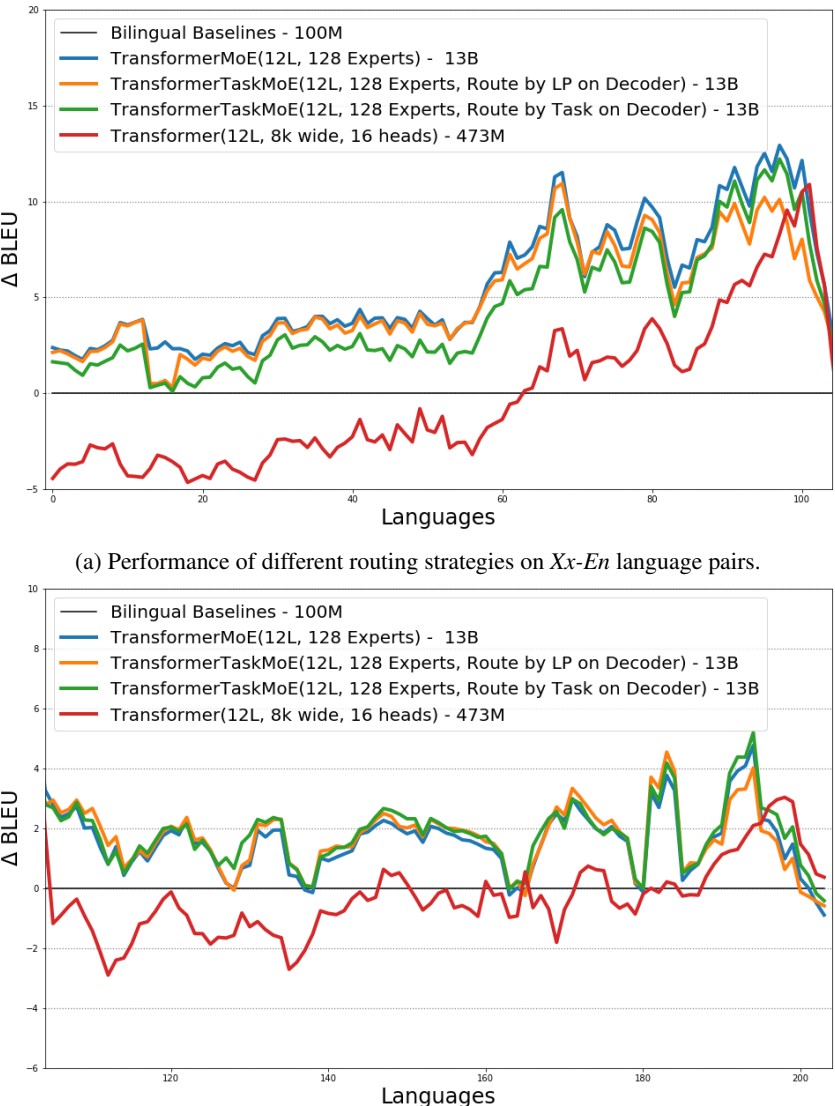

(a) Performance of different routing strategies on *Xx-En* language pairs.

(b) Performance of different routing strategies on *En-Xx* language pairs.

Figure 2: **Comparing the performance of different routing strategies for Mixture-of-Experts (MoE) models on a massively multilingual dataset** – We compare routing experts by tokens, and tasks (using either language pairs or target languages). Given that routing by token on the encoder and routing by task on the decoder performed the best on WMT (Table 1), we use those settings for the scaled up 128 expert models we compare. We split the comparison of results into (a) *Xx-En* language pairs and (b) *En-Xx* language pairs. The languages on the x-axis are sorted left-to-right in descending order of resource size. Best seen in color. Note that the token-level MoE has 6.5B parameters in the decoders while our task-level MoE has only 200M.

to and from English), with a total of 25 billion sentence pairs. This dataset covers a diverse range of domains and languages, and is quite noisy. There is also a heavy imbalance when it comes to the number of examples available per language pair, ranging between $10^4$ and $10^9$ sentence pairs. In order to record gating decisions while controlling for semantics, we created a multi-way aligned evaluation set containing nearly 3k sentence pairs for all languages.[1]

---

[1]Each sentence in our evaluation set is semantically identical across all other languages.

**Model:** We use the 473M Transformer Big (Vaswani et al., 2017) architecture (or modified versions of it in the case of sparse models) as described by Chen et al. (2018) for this set of experiments. Similar to Section 4.1, we (1) share all parameters across language pairs including softmax layer and input/output word embeddings, (2) pre-pend a `<2xx>` token to the source sentence to indicate the target language and (3) use a Sentence Piece Model Kudo & Richardson (2018) with 64k tokens vocabulary shared on both the encoder and decoder side.We followed the training and architecture as shown in Lepikhin et al. (2020).[2]

### 4.3.2   RESULTS

We compare Task-level MoEs and Token-level MoEs to their bilingual and multilingual baselines in Figure 2. We train 128 expert MoE models with routing in these settings: (1) Routing by token on both the encoder and decoder, (2) Routing by token on the encoder and by target language on the decoder and (3) Routing by token on the encoder and by language pair on the decoder.

We find that these scaled up sparse models perform better than their dense baselines, with hybrid task-level routing performing slightly better on *En-Xx* language pairs and pure token-level routing performing slightly better on *Xx-En* language pairs. We hypothesize that for the *Xx-En* tasks, not explicitly dividing expert parameters by tasks on the decoder results in better transfer, thus explaining the better performance of token-level routing. This suggests that a hybrid strategy that partially restricts access to experts based on task-boundaries, while still permitting routing by tokens, might provide the right balance between efficiency and quality.

We also note that while both forms of routing have 13B parameters (6.5B on decoder) at train time, token level routing only on the decoder uses only 200M parameters at inference time, in addition to the practical considerations discussed in Section 3.1. We provide aggregate BLEU scores in Appendix A.6 and parameter count breakdowns in Appendix A.5.

### 4.3.3   COMPARISON OF THROUGHPUT ON MASSIVE MODELS

We further compare Task-level MoEs with Token-level MoEs in terms of throughput across different batch sizes in Figure 4. We measure this by decoding the WMT14 English-German test set with our TaskMoE model and with the baseline TokenMoE model on 128 Cloud TPU V3 cores. We find that our Task-MoE model has 2.6 times higher peak throughput while using 32.34 times less decoder parameters (201M vs 6.5B). Moreover, our Task-MoE model has minimal communication overhead compared to decoding with Token-MoE (0.2% versus 36% of step time).

We measured that the inference time of the token-based MoE model is dominated by the decoder, with the decoders taking 49x the time per step than the encoders. Therefore, the inference cost of task-level routing on decoder only is roughly equivalent to that on both the encoder and decoder.

### 4.3.4   A CLOSER LOOK AT ROUTING DECISIONS

Now, we analyze the routing decisions made in token-level MoE models to further motivate our investigation. We take a token-level MoE model trained on the massively multilingual dataset and decode these models on the multiway tests sets, while logging the routing decisions for every token. We plot the top expert distributions of several tasks with different scripts and language families in Figure 3. For clarity, and because these two groups of languages behave differently in a multilingual setting, we split the gating decisions into those for *Xx-En* and *En-Xx* language pairs.

In the encoder (Figure 3a), tokens from all tasks (*Xx-En*) seem to prefer the same set of few experts slightly over the others. On the other hand, in the decoder (Figure 3b) each task seems to have a slight preference for a few experts over the others. Moreover, the set of experts appears to be similar for related languages. For example, English-Spanish and English-Catalan (two Romance Languages) have similar expert distributions and so do English-Russian and English-Ukranian (two

---

[2]As opposed to displaying BLEU scores for each language pair, we place the baselines on the $x$-axis at zero and report the $\Delta$BLEU trendline of each model we consider. In order to set these bilingual baselines, we train Neural Machine Translation models for each language pair (e.g. a single model for German-to-English), tuned depending on the available training data for that given language We tuned batch-size and different values of regularization methods (e.g. dropout) in a Transformer-Big or Transformer-Base layout, for high or low-resourced languages respectively.

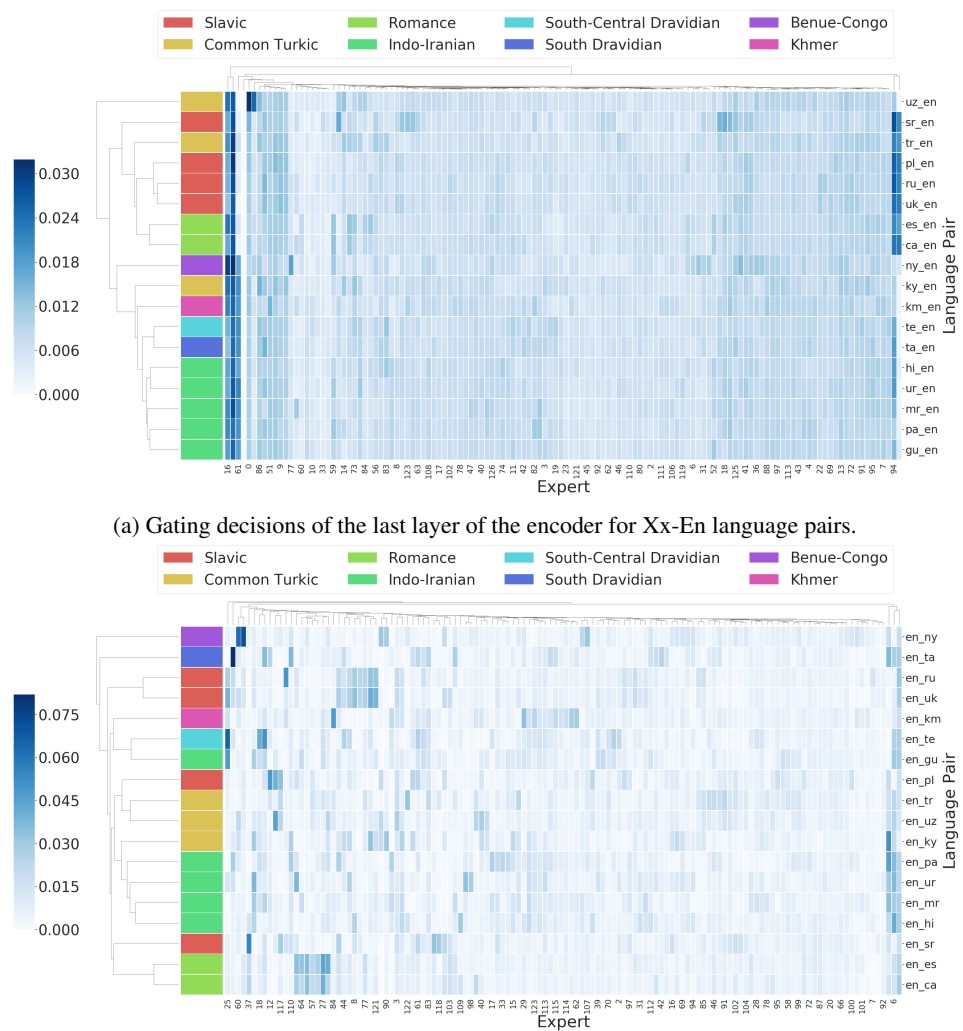

(a) Gating decisions of the last layer of the encoder for Xx-En language pairs.

(b) Gating decisions of the last layer of the decoder for En-Xx language pairs.

Figure 3: We record the gating decisions of our MoE model trained on internal data on a multiway parallel dataset. The darker a cell, corresponding to, say en-sr and the 37th expert, the more the expert is used. In (a) the encoder, tokens from all tasks (*Xx-En*) seem to prefer the same set of few experts slightly over the others; while in (b) the decoder each task (*En-Xx*) seems to slightly prefer a few experts over the other. Moreover, the set of experts appears to be similar for related languages. For example, English-Spanish and English-Catalan (two Romance Languages) have similar expert distributions and so do English-Russian and English-Ukranian (two Slavic Languages).

Slavic Languages). In the Appendix A.7, we provide expert distribution plots for other layers of this model. In addition, we provide expert distributions of the MoE model that routes tokens by target language discussed in Section 2.

Our analysis suggest that, when using token-level routing, task-level decisions emerge naturally in the decoder, providing additional motivation for our proposed routing strategies.

## 5 RELATED WORK

**Conditional Computation:** Conditional computation Bengio et al. (2015), or routing examples through the neural network by activating only a sub-network of the network depending on the input has seen success in large scale natural language processing (NLP) (Shazeer et al. (2017); Lepikhin et al. (2020); Bapna et al. (2019)) and computer vision (Yang et al. (2019)) tasks. A variety of

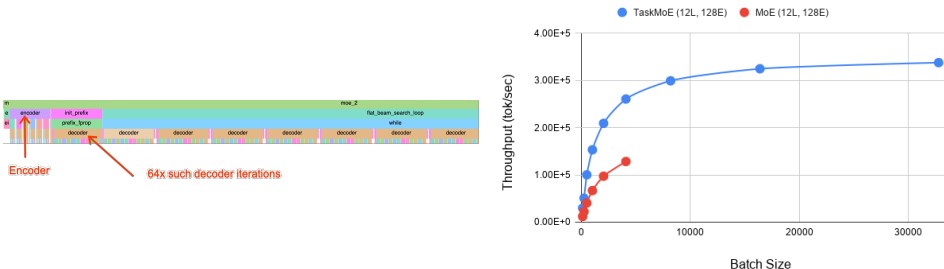

Figure 4: Inference cost analysis. Left, encoders inference cost is small (32ms vs 1565ms) compared to decoders for a 128 Token-MoE model with sequence lenght 64. Right, We measure the throughput of our Task-MoE model and baseline Token-MoE model across batch sizes and see that the peak throughput of Task-MoE is 2.6 times higher. In comparison, the peak throughput of Transformer-Big with 473M parameters is 40.3k tokens/sec.

strategies can be used to route examples such as learning a function on the input Shazeer et al. (2017); Lepikhin et al. (2020), computational budget or Bapna et al. (2019); Elbayad et al. (2019).

**Multi-task Learning:** Multi-task learning Caruana (1997) can improve model performance across all tasks trained on due to regularization and positive transfer between related tasks. Here, sub-networks are be activated depending on the task to which the input belongs - some of these parameters may be shared. This approach has seen success in a variety of domains such as classification, recommender systems and NLP (Ma et al. (2019; 2018); Clark et al. (2019); Collobert & Weston (2008); Ruder et al. (2019); Tan et al. (2019)). Like our work, some of these models have been designed with inference benefits in mind (Ma et al. (2019)). In this work we focus on multi-task learning in the case of multlingual NMT.

**Multi-task learning for Multilingual NMT Models:** Multi-task learning in multilingual models has been well-studied: while complete parameter sharing is simple and works well (Johnson et al. (2017)), an optimal strategy for sharing parameters and possibly having languages-specific parameters would maximize transfer while minimizing interference Hokamp et al. (2019). Strategies involve allocating language specific hidden states, attention modules, decoders or additional specialized layers (Hokamp et al. (2019); Wang et al. (2018); Gu et al. (2018); Bapna et al. (2019)). In addition some strategies involve grouping parameters by language group Fan et al. (2020); Tan et al. (2019). Compared to these works, our approach to parameter sharing is designed to scale models without impacting inference efficiency (as opposed to simply adding language-specific capacity) while still enjoying the benefits of scaling.

## 6 CONCLUSIONS

In this work we discussed more inference friendly algorithms for routing tokens in Sparse Mixture-of-Experts models by making use of task boundaries. We empirically demonstrated that this new algorithm performs as well as, or better than, conventional token-based routing algorithms on two different datasets: the multilingual WMT setup covering 30 language pairs and a large internal dataset covering 200 language pairs. We discussed the trade-offs of these methods in terms of train-time and serving considerations. In addition, we looked more closely at large MoE models and how their gating decisions differ by task.

We conclude by highlighting that the algorithms that are more inference friendly while retaining the training speed advantages of Mixture-of-Experts models are a promising direction for future exploration, motivating research on *inference efficiency* for large models.

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
