# OpenReview forum: "Exploring Routing Strategies for Multilingual Mixture-of-Experts Models"
_ICLR.cc/2021/Conference — Reject_

### Official Review · AnonReviewer1 · 2020-10-26
**Nice observations on task-level routing**

**Rating:** 6
**Confidence:** 5

**Review:**

In this paper, the authors explore alternatives to the standard token-based routing in sparsely-gated MoE models for multilingual NMT. This exploration is motivated by the need for efficient inference in MoE models, for which token-based routing is a limitation. The alternative is task-based routing, where examples for a task are assigned to the same experts. This allows efficient device placement and request dispatch at inference time. The paper compares with the approaches as well as hybrid approaches where different parts of the network use different routing mechanisms. The results show that task level routing is comparable to token-level routing with the added benefit of inference efficiency. Performing task-based routing on the decoder side only gives better better translation quality, at the cost of inference efficiently. An analysis of routing decision in token-based routing justifies the design choices.

Overall, the paper takes a focused problem and experimentally shows an solution that improves inference efficiency. The experiments are well-described and the paper is well-written.
While, the result is interesting it is only a marginal contribution with little novelty in my opinion. It would also be interesting to see how this approaches compares with simpler approaches that deterministically allot parameters to different languages (Wang etal 2018, Bapna etal 2019, Zhang etal 2020)  or language groups (Fan etal 2020).

- Yining Wang, Jiajun Zhang, Feifei Zhai, Jingfang Xu, and Chengqing Zong. Three strategies to improve one-to-many multilingual translation. EMNLP. 2018.
- Ankur Bapna, Naveen Arivazhagan, and Orhan Firat. Simple, scalable adaptation for neural machine translation. EMNLP. 2019.
- Biao Zhang, Philip Williams, Ivan Titov, and Rico Sennrich. Improving massively multilingual neural machine translation and zero-shot translation. ACL. 2020.
- Fan, Angela and Bhosale, Shruti and Schwenk, Holger and Ma, Zhiyi and El-Kishky, Ahmed and Goyal, Siddharth and Baines, Mandeep and Celebi, Onur and Wenzek, Guillaume and Chaudhary, Vishrav and Goyal, Naman and Birch, Tom and Liptchinsky, Vitaliy and Edunov, Sergey and Grave, Edouard and Auli, Michael and Joulin, Armand. Beyond English-Centric Multilingual Machine Translation.  arXiv:2010.11125 preprint. 2020

---

> ### Author Response · Authors · 2020-11-25
> **Response to Reviewer 1**
>
> Thank you for your response.
>
> “compares with simpler approaches that deterministically allot parameters to different languages...”
>
> We have discussed our approach in comparison to the works you have mentioned in the Related Work section in the updated draft of the paper.
>
> In addition to adding quality from allowing the model to learn how to share parameters among tasks, our approach to parameter sharing is designed to scale models without impacting inference efficiency (as opposed to simply adding language-specific capacity), which have not been addressed by the approaches mentioned - we have discussed further results on this in the general response and in the updated draft.
>
> We would also like to note that (Fan et al 2020) is a very recent paper (Oct 21) that was not public at submission time. We also tried to train an MoE baseline with fixed experts, but faced model instability issues.

---

### Official Review · AnonReviewer2 · 2020-10-28
**Review comments for paper 1065**

**Rating:** 4
**Confidence:** 4

**Review:**

This paper compares different routing strategies in Mixture-of-Experts for multilingual NMT, and proposes to route by tasks instead of token, which can achieve better or comparable translation accuracy measured by BLEU and also enable separation of network structures at decoding time with affordable serving costs. The paper claims that with task-level routing, the server only needs to pre-load K experts (assuming top-K routing for MoE layers) during inference, instead of loading all experts as in token/sentence level routing.

Overall, I appreciate the analyses and comprehensive experiment studies in this paper, and also the large-scale in-house datasets used in experiments. The experiment findings about the different routing strategies in the encoder and decoder in en-xx and xx-en settings are also interesting. The connections between the gating distribution and the similar of languages in Figure 3 also make sense.

However, I doubt the novelty and machine learning contribution of this paper: 1) The different routing strategies are natural and seems to have already been proposed by previous works. e.g., for task-level routing, [1] used similar kind of mixture of experts in the language level. 2) This paper simply studies different routing strategies, which is more like empirical analyses. Although the results are somewhat interesting, they are not surprising and most findings are in expectation.

[1] Universal Neural Machine Translation for Extremely Low Resource Languages. https://arxiv.org/pdf/1802.05368.pdf

Meanwhile, I have some questions on the experiment settings: when comparing with the single multilingual base model in Table 1 and Figure 2, the parameters of the MoE model are larger than the single multilingual model (e.g., 533M vs 142M in Table 1). Therefore, it is obvious that MoE model achieves better accuracy than smaller single multilingual model. The MoE models should compare with a single multilingual model with the same amount of parameters.

Another important thing I need to point out is that this paper seems to violate the anonymous policy of ICLR 2021. In Section 4.3.1, the paper says "We use an in-house training corpus (Arivazhagan et al., 2019)".  However, Arivazhagan et al., 2019 shows the authors from Google, which reveals the organization of authors in this paper.  Meta reviewer can further double confirm if this violates the policy.

Besides, this paper is not carefully written and there are many typos which affect the reading. e.g., 1) Two "wo_e" in the line below the equation in Section 2; 2) "to route the token to a select few experts", there is an additional "a"; 3) "a learning rate of a learning rate of 3.0" in Section 4.1.

---

> ### Author Response · Authors · 2020-11-25
> **Response to Reviewer 2**
>
> Thank you for your review - we have addressed the comments on novelty in our general response.
>
> “The MoE models should compare with a single multilingual model with the same amount of parameters.”
>
> We have added a larger 473M Transformer Big baseline to Table 1 with which our method is competitive (baseline is +0.2 BLEU on average). However, we reiterate that our focus is achieving comparable results on scaled models that take similar amounts of train time as its corresponding dense model - in fact, for Figure 2, a 13B parameter dense baseline model would take significantly more time to train.
>
> “...[1] used similar kind of mixture of experts in the language level...”
>
> In [1], the authors used 1 deterministically allocated expert per auxiliary language (task) after the encoder with the purpose of “representing each token in a low-resource language as a context-dependent mixture of the auxiliary language experts”. On the other hand, our method uses learned experts in the feed-forward layer of the transformer to increase the scale of the model with minimal added cost at both train time and inference time.
>
> “Another important thing I need to point out..”
>
> Thank you for flagging the issue - we have addressed this in our updated draft and will accept the decision of the Meta-Reviewer.
>
> “there are many typos…”
>
> We have corrected these typos and more in the updated draft.

---

### Official Review · AnonReviewer4 · 2020-10-29
**A mixture model for unified multilingual neural machine translation. Though idea is simple and straightforward. The novelty is limited and the experiments are not satisfying.**

**Rating:** 5
**Confidence:** 4

**Review:**

This paper introduces several routing strategies for multilingual neural machine translation. The motivation is to train a single mixture model that can serve the training and prediction of multiple models. Specifically, several strategies are proposed: token-level, sentence-level and task-level. Experiments on WMT and massive multilingual NMT dataset show that the proposed approach outperforms the vanilla unified multilingual model. While this approach is simple and straightforward, I have some concerns.

Pros:
- A mixture model is proposed for multilingual NMT, forming a hierarchical structure:  token-level, sentence-level, and task-level.
- The general framework is reasonable, straightforward, and easy to implement.
- Experiments are extensive (both the WMT data and massive NMT data).


Cons:
- The general idea is not novel. Building a mixture model for multi-task learning has been well studied in the literature [1,2] (not cited). The relation may need to be clarified.
- In the experiment part, the results of single models (not unified multilingual model) need to be reported.
- Comparisons with other unified multilingual approaches are required [3,4],



[1] Ma et al. Modeling Task Relationships in Multi-task Learning with Multi-gate Mixture-of-Experts, KDD 2020

[2] Ma et al. Snr: Sub-network routing for flexible parameter sharing in multi-task learning.

[3] Tan et al. Multilingual Neural Machine Translation with Language Clustering.

[4] Multilingual Neural Machine Translation with Knowledge Distillation.

---

> ### Author Response · Authors · 2020-11-25
> **Response to Reviewer 4**
>
> Thank you for your review. We would like to clarify that our approach scales models without significantly impacting inference efficiency and have added more measurements in our updated draft of the paper and general response to highlight the same.
>
> “the results of single models (not unified multilingual model) need to be reported.”
>
> We have updated the draft with comparisons to bilingual models, and our method outperforms bilingual baselines by 2.5 BLEU on average.
>
> We included a discussion of the related works pointed out in the the updated draft:
>
> “The relation may need to be clarified…”
>
> In [1], the authors introduce a gating function for each task, whereas we use simpler gating functions which we show to be more inference friendly. Compared to [2], our method has a guaranteed inference cost and decreases the serving cost by a significantly larger fraction.
>
> “Comparisons with other unified multilingual approaches are required [3,4]”
>
> In [3], the authors share parameters by training different models according to predetermined language clusters while our method learns which parameters to share within a unified model, while in [4] the authors take multiple bilingual models and distill them into a single multilingual model -  our model does not require a distillation step and we see these methods as not comparable.

---

### Author Response · Authors · 2020-11-25
**General Response to Reviewers**

We thank reviewers for the comments and feedback. We first address general concerns among reviewers.

1. Novelty
Despite sharing similarities with existing works, we believe ours is the first work that performs comprehensive studies of routing strategies for sparsely-gated mixture-of-experts models. Existing applications of sparsely-gated mixture-of-experts models in NLP, such as https://arxiv.org/abs/1701.06538, have generally made routing decisions only at the token level.

Another important aspect of our work is to demonstrate that task-level sparsely-gated mixture-of-experts can lead to significant improvements in terms of inference efficiency while being competitive in translation quality. To highlight the inference efficiency benefits of our approach, we have added additional measurements to the updated draft. We assume under the ideal case where the serving batch size can be as large as possible, we demonstrated that the task-level MoE approach can still achieve upto 2.6x higher throughput and negligible communication overhead (vs 36% of time being spent in communication). Moreover the task-level decoder has 32.3x less parameters at inference time, which in turn required much less accelerators to load the model. This is especially useful for the more practical cases when the server can only accumulate smaller batch size.


2. Experimental results
In response to feedback we have added some additional baselines to our draft in Table 1: (1) Bilingual baselines (2) A larger 473M Transformer Big baseline. We also tried to train an MoE baseline with fixed experts, but faced model instability issues - suggesting that there may be benefits to learned experts.

Our method outperforms bilingual baselines by 2.5 BLEU. On the other hand, our method is competitive with the dense Transformer Big baseline (Transformer Big baseline is +0.2 BLEU on average).  Note that TransformerBig incurs much higher decoding cost. We measured our task-level MoE achieved 8.4x higher decoding throughput. Moreover, we reiterate that the motivation behind scaling sparsely is to increase capacity with little overhead while remaining competitive with dense models - for example, while it is feasible to train a 473M parameter model, training a dense 13B model (the size of our scaled up model in Figure 2) is prohibitively slow and expensive.

We have updated our draft to reflect these changes.

---

### Decision · Program_Chairs · 2021-01-07
**Final Decision**

**Decision:**

Reject

**Comment:**

This paper proposes routing strategies for multilingual NMT. The motivation is to train a single mixture model that can serve the training and prediction of multiple models. Several strategies are proposed: token-level, sentence-level and task-level. This is a simple and straightforward approach (which is fine). The main concerns from the reviewers regard novelty and missing comparisons. In their updated draft, the authors added comparisons to bilingual models and they added a discussion wrt related work. However, the author’s response did not address enough some of other reviewers’ concerns regarding comparison with other approaches, and the lack of novelty persists (mixture models for multi-task learning have been previously proposed in the literature), which makes me lean towards rejection. I suggest the authors address these aspects in future iterations of their work.